# View from the Biological Property: Insight into the Functional Diversity and Complexity of the Gut Mucus

**DOI:** 10.3390/ijms24044227

**Published:** 2023-02-20

**Authors:** Chengwei He, Han Gao, Shuzi Xin, Rongxuan Hua, Xueran Guo, Yimin Han, Hongwei Shang, Jingdong Xu

**Affiliations:** 1Department of Physiology and Pathophysiology, School of Basic Medical Sciences, Capital Medical University, Beijing 100069, China; 2Department of Clinical Medicine, School of Basic Medical Sciences, Capital Medical University, Beijing 100069, China; 3Department of Oral Medicine, School of Basic Medical Sciences, Capital Medical University, Beijing 100069, China; 4Experimental Center for Morphological Research Platform, Capital Medical University, Beijing 100069, China

**Keywords:** mucus, mucin, phenotypic switch, biological characteristics, barrier function, health and illness

## Abstract

**Highlights:**

**What are the main findings?**
Mucus, as the most widely distributed biofilm and vital protective barrier on the surface of mucous membranes throughout the body, fulfills a number of critical activities in the maintenance of cellular and organismal homeostasis.Morphologic and biochemical evidence corroborate that the mucin family is composed of the transmembrane and gel-forming mucins, which have diverse functional modules.

**What is the implication of the main findings?**
Goblet cells are responsible for synthesis, storage, and secreting, which are implicated in either beneficial or detrimental factors.Excessive or insufficient mucus secretion as well as phenotypic alternation may be associated with gastrointestinal disorders, offering potential therapeutic targets for prevention in clinical practice.

**Abstract:**

Due to mucin’s important protective effect on epithelial tissue, it has garnered extensive attention. The role played by mucus in the digestive tract is undeniable. On the one hand, mucus forms “biofilm” structures that insulate harmful substances from direct contact with epithelial cells. On the other hand, a variety of immune molecules in mucus play a crucial role in the immune regulation of the digestive tract. Due to the enormous number of microorganisms in the gut, the biological properties of mucus and its protective actions are more complicated. Numerous pieces of research have hinted that the aberrant expression of intestinal mucus is closely related to impaired intestinal function. Therefore, this purposeful review aims to provide the highlights of the biological characteristics and functional categorization of mucus synthesis and secretion. In addition, we highlight a variety of the regulatory factors for mucus. Most importantly, we also summarize some of the changes and possible molecular mechanisms of mucus during certain disease processes. All these are beneficial to clinical practice, diagnosis, and treatment and can provide some potential theoretical bases. Admittedly, there are still some deficiencies or contradictory results in the current research on mucus, but none of this diminishes the importance of mucus in protective impacts.

## 1. Introduction

The gastrointestinal (GI) tract is a complex set of organs responsible for the crucial roles of food ingestion and digestion, nutrients’ absorption, and waste evacuation, which maintains its structural integrity by preventing digestive enzymes and luminal pH from being compromised under physiological conditions [1]. The stability of the structures is necessary for their function, as depicted in Figure 1, and, as a result, the epithelia must retain their structural and functional integrity in the context of the appropriate living circumstances. Understanding how to protect the organism from invading pathogens and harmful substances has attracted considerable critical attention. Extensive experimental and observational evidence corroborated that the complex physical barrier, consisting of a monolayer of polarized epithelial cells and an underlying immunocyte, protects against exposure to the potentially gastrointestinal pathogenic flora and so preserves homeostasis [2,3]. The human digestive tract is the largest and most complex organ of microbial accumulation. The homeostasis of the intestinal milieu and metabolic products of microorganisms have a critical mission in maintaining body health [4,5]. The concept that epithelial barriers are not only particularly vulnerable, due to their exposure to a variety of symbiotic and pathogenic as well as harmful substances, but also serve as barometer of physical health is generally accepted [6,7]. Recent advancements in barrier integrity have led to a proliferation of studies on the correlations between intestinal mucus and various diseases. The characterization of mucus is critical for gaining a better understanding of multiple functions [8]. Despite this extraordinary level of attention, there is an urgent need to address the underlying mechanism associated with mucin-related disease. Therefore, the summary of the many examinations of the mucus structure and organization in this study offers valuable insight into the crucial influence of mucus on health. Furthermore, we also sum up additional underlying mechanisms linked to an increased risk of disease and aberrant mucin. We could not help but mention the limitations of the existing mucus studies based on the available data. Beyond the aforementioned aspects, the goals of this summary are to highlight the integrity of the mucus layer regarding the origin of health and find specific associations between certain kinds of mucins and diseases, to offer research directions for new therapeutic targets or a novel gut-related disease biomarker.

## 2. Mucus Biological Properties

The surface mucosal layer of the GI is composed of two known cell types: absorbable cells and secretable cells, which are further subdivided depending on their visual histological characteristics and depicted in Box 1. Emerging evidence demonstrates that the mucus throughout the gastrointestinal tract contains the same biological components, but the mucus characteristics differ by area. Mucus is essential for maintaining health, and its properties can be influenced by a variety of factors.

Box 1Brief review of the differences between absorbable nuclei and secretory cells.Wrinkles or folds structurally cover absorbable cells. Microvilli are finger-like projections on individual epithelial cells. The plicae circulars, villi, and microvilli serve to
enhance the amount of surface area that is accessible for nutritional
absorption. Secretable cells can be characterized, in addition to their
glandular form, by the manner of secretion and the type of chemicals emitted.
These secretions are encased in vesicles that migrate to the cell’s apical
surface, where they are released by exocytosis. Saliva containing the
glycoprotein mucin, for example, is a merocrine secretion.

### 2.1. General Biological Characteristics of Mucus

The term mucous membrane derives from the fact that mucus is the principal material released, and the primary component of mucus is a mucopolysaccharide, which is also named mucin. The mucus embodies a multitude of concepts, including mucous membranes, extensively covering the cavities and canals connected to the exterior, principally the respiratory, digestive, and urogenital tracts, as shown in Figure 1. Mucous membrane structures vary, but they all consist of an epithelial cell layer and connective tissue layer. A complex viscoelastic mucus, released by the goblet epithelial cells and mucinous cells present in epithelial tissue, is composed of numerous components: water (90–95%), lipids (1–2%), electrolytes, proteins, and others [9,10] that are closely tied to its structure and function. The amount of electrolytic mass varies considerably across the tissues and may deeply impact the hydration and rheology of mucus. To maintain the wetting, hydrophobicity, and barrier function of the mucous layer, a particular number of lipids is required in mucus. Proteins and immunoglobulins act as a dynamic semipermeable barrier between the epithelium and the lumen contents, providing a protective layer of immune function by effectively recognizing various invading pathogens, activating the immune response to relieve the entry of pathogenic microorganisms, and eliminating their negative consequences. Thus, when microbial colonies in the GI tract gradually develop, the thickness of the mucus layer likewise increases from the upper GI tract to the lower GI tract, reaching its thickest in the colon: from 200 to 800 μm, depending on the morphology [11]. Although a single, permeable, rapidly self-renewing interface forms in the small intestine, antimicrobial agents keep pathogens away from its primary site of action. Within the colon, on the other hand, a double layer of mucus forms, with the inner layer firmly adhering to the epithelial cells to provide protection and the outer layer expanding into a loose outer layer to serve as a habitat for commensal bacteria [12]. The normal epithelial layer of the membrane consists of the squamous epithelium, which is often located in the mucous membrane of the skin, but a simple columnar epithelium differs significantly from this and instead resembles a columnar epithelium. The cells, which possess the peculiarity of fast self-renewal and a certain toughness to withstand the wear and tear associated with food particles and other forms of invasion of various digestive enzymes, pathogenic microorganism, and toxins, are significantly greater in height than width and are typically scattered among the distal gastrointestinal tract and respiratory tract.

### 2.2. Mucus Serves as a Crucial Entryway to the Body’s Barrier

Due to the rapid rate of mucus replacement, goblet cells constantly secrete mucin to complement and retain the completeness of the mucus barrier; however, goblet cells are impeded by multiple adverse factors, including microbes, toxins, and cytokines, leading to a progressive loss of function, which can also harm the integrity of the mucus barrier upon altering its phenotype.

#### 2.2.1. Physical Barrier Function of Mucus

Data from a variety of studies on thicknesses have confirmed that several hundred micrometers becomes responsible not only for an indispensable barrier between the lumen milieu and the host tissue but also as a habitat for microorganisms. There is also a transmembrane mucus barrier, which is generated mostly by the glycocalyx tethered to the cell membrane of the colon epithelium and has a significant influence on a strongly cross-linked inner mucinous layer made of mucin 2 (MUC2). The outermost layer is a loose outer mucus layer that forms following mucin hydrolysis in the inner mucinous layer.

From an ecological standpoint, aside from serving as a niche carbon source and providing nutrients, mucus glycans include a range of bioactive compounds and can function as binding sites for bacteria, to prevent them from accessing the inner layer of mucus in a common condition [13]. Mucin’s oligosaccharide side chains attach to the adhesins utilized for bacterial colonization, functioning as bait to capture numerous fungal and bacterial pathogens and prevent them from contacting and translocating through epithelial cells into the underlying stroma, where they might provoke an immune response. In vivo studies showed that the mucin’s imperfections might lead to the accumulation of additional pathogens on the mucosal surface. Notably, while the total number of infected bacteria may seem modest, such alterations can cause severe illness or even death [14]. One possible explanation is that a lack of mucus causes a reduction in some of the symbiotic bacteria that rely on mucus as a carbon source, which decline due to an insufficient nutrient intake. Another possible explanation is that the lack of MUC2 makes the intestinal tract more vulnerable to bacterial infection and has a robust immune response. These in vivo results’ analyses are essentially compatible with the in vitro data obtained from the *Muc2*^−/−^ mice model of spontaneous colitis and colon cancer. An MUC2 deficiency resulted in the colonization of different bacterial communities in the intestinal tract, and the abundance of *Ruminococcaceae* and butyric-producing bacteria were significantly increased to a certain extent [15]. Moreover, recent evidence has demonstrated that mucopolysaccharides contribute to the preservation of the integrity of the intestinal barrier by forming a thick gel layer [16], along with hydrodynamic lubrication [17], and may also be influenced by the properties of intestinal peristalsis [16].

#### 2.2.2. Immune Barrier Function of Mucus

Mucus, in addition to mucin, contains many components that have an impact on gut flora; the mucin network and water establish a diffusion barrier that maintains a healthy local innate defense system, thereby performing multitudinous functions under physiological as well as pathophysiological conditions. Many pathogens infiltrate the host through mucosal surfaces, and immunoglobulin A (IgA) is regarded as the principal antibody homotype responsible for protection at these sites. IgA helps the host defense by neutralizing bacterial toxins, food antigens, and viruses as well as inhibiting bacterial adhesion and movement [18]. Recurrent respiratory infections are the most prevalent IgA deficiency linked to increased infections within the mucosa-associated lymphoid tissue (MALT) [19]. In the gut, upon bacteria csis, B cells undergo a class switch to IgA^+^-secreting plasma cells (PC-Ac) in mucosal-associated lymphoid tissues. Notably, germ-free (GF) mice have deficiencies in numerous particular immunocyte populations, such as specialized goblet cells and IgA-producing plasma cells; this renders them more vulnerable to immunological infection, and, in general, creates greater susceptibility to immune infection. In contrast, recurrent respiratory infections are the most prevalent symptom of IgA deficiency, affecting approximately 20–30% of IgA-deficient individuals [20], which may be due to compensatory increases in secretory IgM [21] or low levels of mucosal IgA that adequately perform a protective function against mucosal infections [22]. Secretions are most typically affected by distinct mucosal locations, with their relative proportions larger in the duodenum and ileum mucus and smaller in the colon mucus. The above description presents a snapshot of the protective role of the several anti-bacterial or viral infection effects of serum IgA, proposing a protective role through neutralization or complement activation by an interaction with the IgA-specific receptor FcαRI. Additional studies indicated that reduced or aberrant IgA lacking an IgA-translocated polymeric Ig receptor (PigR) still has mucus to separate myxobacteriosis from the epithelial cells in the colon. Another, paneth cells, highly specialized AMP-secreting epithelial cells distributed at the base of the crypt of the small intestine, produce not only defensin and lysozyme such as cathelicidin (LL37), α-defensin, lysozyme, and RegIIIβ/γ but also MUC2, which mixes with the antibacterial peptides; the epithelium is covered with a bilayer of mucus in the colon, whereas the inner mucus layer restricts bacterial access to the epithelium and the loose outer mucus layer as niches for the commensal bacteria that provides a rich source of nutrients [23,24]. Additionally, mucus also consists of other large molecules that are secreted by goblet cells, such as FCGBP, CLCA1, ZG16, AGR2, TFF3, and KLK1, on which little research has been conducted. Despite considerable research growing up around the theme of mucus, the immunological roles or other functions of these molecular mechanisms that underpin immune barrier function remain inadequately discussed.

The viscoelastic secretion is produced by goblet or mucous-producing cells and is found on the epithelial surfaces of all organs accessible to the outside world. Mucus is a complex aqueous fluid having viscoelastic, lubricating, and hydrating qualities due to the glycoprotein mucin in conjunction with electrolytes, lipids, and other smaller proteins, under physiological circumstances illustrated in Box 2. However, mucin secreted under pathological conditions has multiple structural differences, and its limitations affect the normal physiological role of mucus, resulting in some disease-related changes, such as an alteration in its dense cellular permeability, which allows pathogenic microorganisms and poisons to directly stimulate the epithelial cells. A great number of studies showed that the immune escape reactions of tumor cells are common in tumors when the structure of the mucus changes.

Box 2Concept of mucin and mucus.Mucins are a major family of slimy
glycoproteins found in practically all organisms. Glycoproteins are a class
of proteins that have carbohydrate groups bonded by a polypeptide chain and
are a prominent component of mucus. A mucus is a gel made up of water, ions,
proteins, and macromolecules. The mucin glycoproteins are key components that
are crucial for the local protection of the underlying epithelium from
harmful stimuli.

## 3. Mucin Glycoproteins as the Main Component of Mucus

Highly glycosylated mucin, as the dominant structural macromolecular component of mucus, has a unique tandem repeat sequence of amino acids that are rich in serine, threonine, and proline sequences, also known as the PTS-rich region. The other regions are non-repeating, with a carboxyl and amino terminus, which may be rich in Cys residues without much Ser/Thr. This has N- and O-glycosylation, which are uncommon [9,10], as the principal site for the O-linked glycosylation of the molecule. Human mucins are encoded by at least 21 unique mucin genes (MUC), which are split into two families: secreted and membrane-associated. Goblet cells become activated in response to a variety of stimuli, including cytokines, proteases, bacteria, and hazardous pollutants; this occurs at either the transcriptional or post-transcriptional level, mediated by different intracellular cascades. Due to accumulating evidence linking the mucin to some diseases, the role of mucin has attracted much attention for its functional significance. Here, we emphasize insights into the fundamental mechanism of the mucin synthesized in goblet cells, which may shed light on future clinical practice [25]. 

### 3.1. Synthesis and Secretion of Mucin in Goblet Cell

It is now widely acknowledged that there are two types of mucin, including secreted and membrane-associated [25]. Mucin is mainly composed of a single, highly O-glycosylated protein that is synthesized, stored, and secreted by highly polarized exocrine goblet cells in a TLR- and NLRP6-dependent manner. Mucin is the main secreted glycosylated protein that occupies the main spatial distribution in goblet cells, which are recognized by their apical accumulation of glycoproteins’ storage granules. Through collaboration, Wnt/Notch signaling controls the process of the proliferation, fate, differentiation, and death of the metazoans’ intestinal crypt cells. Some experimental evidence confirmed that potentiating PI3K-AKT signaling can boost luminal cell proliferation, suppress the anoikis of the luminal epithelial cells by enhancing NF-B activity, independent of Hes1, and then restore the capacity of the luminal progenitors for unipotent differentiation and short-term self-renewal in vivo and in vitro [26]. While the colonic epithelial stem cells located in the crypt bottom migrate toward the top of the crypt and differentiate into goblet cells through boosted Wnt signaling or Math1 (encoded by the gene *Atoh1*, also called Math1) activation [27,28], Krüppel-like transcription factor 4 (Klf4), SAM pointed domain-containing ETS transcription factor (Spdef), and growth factor independence 1(Gfi1) are three of the proteins that drive the goblet cells’ differentiation in the colon [29,30].

The integrity of the mucus layer is critical for health. Mucin is a major component of mucus, and its synthesis and secretion are regulated by multiple factors. Therefore, the regulatory pathways underlying its physiological condition have been attracting attention, since it is a hotly pursued therapeutic target in clinical practice. Mucin synthesis, as shown in Figure 2C, like that of many other secretory glycoproteins, is dominated by the perinuclear localized rough endoplasmic reticulum (ER), which involves the linkage of distinct saccharides. After the peptide core of the mucin was transferred to the ER, the folding of the cysteine-rich N- and C-terminal domains and the formation of disulfide bonds, as well as N-glycosylation and C-mannosylation, continued in the Golgi apparatus. This is essential for the dimerization of mucins and their transport from the ER to the Golgi apparatus. Mucins are then transported through vesicles to the cis-Golgi apparatus for further modification. GalNAc initializes the O-glycosylation of serine and threonine in the presence of enzymes in the cis-Golgi apparatus. The Golgi compartments are the primary reprocessing site for synthesis, assembly, and secretion, in which the O-oligosaccharides further form linear or branching core structures by connecting monosaccharides, Gal or GlcNAc, to multiple locations of the initial GalNAc, and, on the basis of this basic skeleton, alternate linking to extend O-linked oligosaccharides. In the trans-Golgi apparatus, termination of the O-link is achieved by adding fucose, GalNAc, or NeuNAc in a competitive enzyme. Synchronously, terminal sialic acid and sulfate provide a considerable quantity of negative charge to the mucin oligosaccharides, causing the oligosaccharide chains to repel each other and form extended rod-like shapes. After synthesizing and being bound by the disulfide bonds in the amino-terminal vWFD domain to form higher-level mucin polymers, mature mucin turns into secreted granules. Many negative charges on the mucin molecules are neutralized by positive ions, such as H^+^, Na^+^, Ca^2+^, etc., during this process, making the volume of mucin fluctuate substantially before and after secretion. The mature secretory granules go through a series of signaling processes before moving to the cell’s outer apical area and being released on a predefined timetable [31].

The Golgi complex may aid in the synthesis, assembly, and secretion of O-oligosaccharides. Higher-level mucin polymers preferentially develop into secretory granules after being packed in vesicles and being bound by disulfide-bonded dimers in the amino-terminal vWFD domain. Indeed, many of these negative charges on mucin molecules are neutralized by positive ions, such as H^+^, Na^+^, and Ca^2+^, during this process, resulting in a volume change undoubtedly and dramatically associated with the changes in elasticity before and after secretion. Essentially, these immature secretory granules migrate along the cellular secretory pathway from the ER to the Golgi complex, where they mature into a secretable particle. Rab is primarily responsible for the granule-to-actin transition of mature secretory granules along microtubules The secretion of mucus vesicle particles is, undeniably, also regulated though other factors, such as myristoylated alanine-rich C-kinase substrate (MARCKS) protein, Syntaxin, Munc13, Munc18, SNAP2, diacylglycerol (DAG), and inositol triphosphate (IP_3_); the detailed information is depicted in Figure 2, which occurs when the heptahelical receptors on the cell membranes are activated by the stimulatory signals that promote mucin secretion including ATP, acetylcholine, and histamine. DAG activates MUNC-13, allowing it to open syntaxin with Munc18, which forms a four-helix bundle SNARE complex with SNAP23 and VAMP. IP_3_ stimulates Ca^2+^ release from the ER, hence activating the calcium sensor synaptotagmin and causing additional SNARE complex shrinkage. Previous experimental evidence shows that secretory granules move along a concentration gradient owing to diffusiophoresis in the presence of solute anions and cations, implying that the compressed mucin may fail to expand outside the cell due to a lack of charge shield, as explained in [32,33,34].

### 3.2. Transmembrane Mucins

Membrane-bound mucins, also widely termed transmembrane mucins (TM), which localize to the apical surfaces of epithelial cells and gel mucin, are in contact with relatively harsh environments. The TM family consists of 12 members, including MUC 1, 3, 4, 11, 12, 13, 15, 16, 17, 20, 21, and 22, which differ in their extracellular length, domain architecture segment, and cytoplasmic signaling domain. An overview of the detailed information is presented in Table 1 and Table 2. Its functions and relationship with disease are discussed in the following sections.

As shown in Figure 2B, the extracellular domain of TM contains a varying quantity of the tandem-repeat (TR) domain; the sperm protein, enterokinase, and agrin (SEA) domain; or the epidermal growth factor (EGF)-like domain. The centrally located TR regions are the distinguishing property of all mucins, differentiating them from other membrane-bound glycoproteins, although a wide range of TM glycoproteins have a particular trait distinguishing them from other TM glycoproteins. 

Most TM members have a highly GalNAc-O-glycosylation extracellular domain, which is connected to the extracellular apex via a long N-terminal mucin domain that has important barrier functions and a short cytoplasmic C-terminal domain that can be phosphorylated upon activation of the intracellular signal transduction pathways, as illustrated in Figure 2B. Different functional regulatory peptides, such as the sea urchin sperm protein, enterokinase, and agrin (SEA) and epidermal growth factor (EGF)-like domains, are implicated in various types of TM [35]. The most striking observation obtained from the preliminary analysis of the biochemical structure via N-terminal sequencing and multidimensional NMR spectroscopy revealed that MUC1, MUC3, MUC12, MUC13, and MUC17 are coupled to the cell membrane via the SEA domain, and MUC4 is coupled to the cell membrane via the vWD domain [36]. MUC17, as previously indicated, is strongly expressed on the intestinal epithelium surface and was originally identified as a highly associated gene with an unclear underlying mechanism. Overexpression of MUC17 in a 2D Caco-2 cell model system lead to a reduction in the *Escherichia coli* binding to the cell surface via a TNF-dependent manner, as did overexpression of MUC1 against *Helicobacter pylori* infection [37]. Equally important, dynamic changes in MUC1 are involved in the generation and development of a wide variety of tumors, which may become a viable diagnostic and targeted therapy indicator in future clinical practice [38,39,40].

### 3.3. Secreted Mucins

Similarly, the most essential component of the mucus family is secreted mucins, which can be further subclassified as gel-forming mucins MUC 2, 5AC, 5B, 6, and 19 and tiny soluble mucins MUC 7, 8, and 9. Besides the main detailed overview in Table 1 and Table 2, additional instructions are available in Box 3. Structurally, the gel-forming mucins are flanked by the C- and N-terminus to form a dimer, which, together with the other parts, forms a larger polymer and eventually forms a gel mucus. These O-glycosylated protein molecules are linked to each other by disulfide bonds between the Cys domain to form giant molecules that mix and expand to produce mucus, once they are discharged from the goblet cell into the lumen. Since extensive O-glycosylation protects the core mucin domains that create gel mucins from being broken down by proteases in the gut, they form mucus to protect and lubricate the gut [36]. MUC2 is the main gel -forming mucin in the intestine, which has a mass of about 2.5 MDa and consists of 20% protein and 80% glycans. As described in Figure 2A, MUC2 polymerizes in the colon to form large mesh sheets that stack together like bricks to form an internal mucous layer that does not allow bacteria to pass through. The inner mucus layer in the colon renews about every 1-2 hours [41]. The outer mucus layer has a larger volume than the inner mucinous layer, due to not only the host’s proteolytic activity but also the commensal bacterial proteases and glycosidases, which may allow bacteria to colonize. Previous studies have corroborated that crypt stem cells produce highly proliferative and amplifying daughter cells, depending on the stimulation from the niche, and that they can differentiate into specialized all-epithelial-cell populations within the goblet cells responsible for producing and secreting mucin proteins into the lumen. The thickness of the mucous layer structurally correlates with the number of goblet cells from the small intestine to the colon and the diversity and richness of the intestinal flora functions inside the intestinal goblet cells to govern mucus production [42].

**Table 1 ijms-24-04227-t001:** Summary of classification and biological characteristics of mucin.

Mucin Type		Mucin/Cytogenetic Band	Cell Type Expression	Protein Domains	Number of Amino Acids	Number (Estimated Length) of Mucin Domains	Section of Gastrointestinal Tract	Functions	Refs.
Secreted mucin	Gel mucin	MUC2/11p15.5	Goblet cells Paneth cells	4 VWD, 2 CysD, 1 CK	~5200	2 (~550 nm)	Small intestine, large intestine	Protection, lubrication, entrapment	[43,44,45,46]
		MUC5AC/11p15.5	Mucous cells	4 VWD, 11 CysD, 1 CK	>5050	11 (>350 nm)	Stomach	Protection, lubrication, entrapment	[47,48]
		MUC5B/11p15.5	Mucous cells Goblet cells	4 VWD, 7 CysD, 1 CK	~5700	7 (~550 nm)	Mouth, large intestine	Protection, lubrication, entrapment	[49,50]
		MUC6/11p15.5	Mucous cells	1 VWD, 1 CK	~2400	1 (~250 nm)	Stomach, small intestine	Protection, lubrication, entrapment	[51,52,53]
		MUC19/12q12	Mucous cells	1 VWC	>7000	1	Salivary gland, testis	Protection, lubrication	[54,55]
	Small soluble mucin	MUC7/4q13.3	Mucous cells	None	377	1 (~230 nm)	Mouth	Protection	[56,57,58]
		MUC8/12q24.33	Epithelial cells	None	2699	None	Airway	Protection	[59]
		MUC9/1p13.2	Epithelial cells	None	654	1	Oviduct	Fertilization related	[60,61,62]
Transmembrane mucin		MUC1/1q22	Epithelial cells	1 SEA	~1250	1 (~200 nm)	Mouth, stomach, small intestine, large intestine	Signaling, protection	[63,64,65]
		MUC3II/7q22	Enterocytes	1 SEA	>2550	1 (>350 nm)	Small intestine, large intestine	Apical surface protection	[66,67,68]
		MUC4/3q29	Epithelial cells goblet cells	1 NIDO, 1 AMOP, 1 VWD	~5300	1 (~800 nm)	Small intestine, large intestine	Signaling, protection	[43,69,70]
		MUC12/7q22.1	Enterocytes	1 SEA	~5500	1 (~1000 nm)	Small intestine, large intestine	Apical surface protection	[71]
		MUC13/3q21.2	Enterocytes	1 SEA	512	1 (~30 nm)	Small intestine, large intestine	Apical surface protection	[72]
		MUC15/11p14.3	Gland cells	None	~334	1	Epididymis, thyroid	Antimicrobial activity	[73]
		MUC16/19p13.2	Epithelial cells	33 SEA	~22,000	1 (~2400 nm)	Mouth	Apical surface protection	[74]
		MUC177q22.1	Enterocytes	1 SEA	~4500	1 (~800 nm)	Small intestine, large intestine	Apical surface protection	[75,76]
		MUC20/3q29	Epithelial cells	None	~709	2	Esophagus, lung, stomach, kidney	Signaling, protection	[77,78]
		MUC21/6p21.33	Epithelial cells	None	566		Mouth, stomach, eyes	Signaling, protection	[79,80]
		MUC22/6p21.33	Epithelial cells	None	~1773	1	Esophagus, vagina, lung	Protection	[81,82]

**Table 2 ijms-24-04227-t002:** Comparison of the general characteristics of acidic and neutral mucin.

	Acidic Mucin	Neutral Mucin	Refs.
Main amino acids	Proline, threonine, glycine	Serine, aspartate, alanine	[83]
Main glycosylation	Sialic acid, N-acetylgalactosamine	Fucose, galactose, N-acetylglucosamine	[83]
Location	Stomach:	Surface epithelium, foveolar cells, most of the mucous neck cells	[84,85]
	Small intestine:	Glycocalyx of the brush border, goblet cells of both villi and crypts, especially in distal ileum	
	Large intestine:	All brush border and goblet cells	
Secernent	Bacterial colonization: *Bifidobacterium dentium*, *Helicobacter pylori* Cancer: gastric cancer Drugs, cytokines and chemicals: *Moringa oleifera* leaf powder, keratinocyte growth factor, trefoil peptide, dietary resistant starch type 3	Bacterial colonization: *Salmonella typhimurium* Drugs: butyrate, methotrexate Others: *Nippostrongylus brasiliensis* infection	[86,87,88,89,90,91,92,93,94,95,96,97,98,99]
Antisecretory	Drugs, and chemicals: aspirin, sesame oil	Bacterial colonization: *Helicobacter pylori* Cancer: gastric cancer Others: food restriction, microgravity, aging, vagotomy	[98,99,100,101,102,103,104,105]

Box 3Alkaline mucin.Alkaline mucin is a thick fluid
produced by animals that serves a protective function and confers tissue
protection in an acidic environment, such as in the stomach, and that has
been shown to possess bactericidal properties to protect the cervix and
uterus from microbes. Alkaline mucus might also be produced by the
histological goblet cells of the cystic fibrosis (CF) intestine and the
Brunner gland of the duodenal epithelium [106,107,108]. Its underlying
mechanism includes the activation of carbonic anhydrase (CAII), the cystic
fibrosis transmembrane conductance regulator (CFTR), and the Na^+^/H^+^
exchanger (NHE1) party via the prostaglandins’ secretion of alkaline
solution. However, the details of the mechanism remain unclear.

## 4. Regulation of Mucin Function

Mucin regulation and function in normal and abnormal conditions need an analysis of the MUC-specific proteins. However, the size and complexity of the mucin oligosaccharide structures obstruct the illumination of the MUC-specific underlying processes. As previously mentioned, the synthesis and secretion of the mucin’s pathway is an intriguing player in many biological processes, implying that the regulatory mechanism is complicated and an unresolved mystery. Due to the large amount of experimental evidence, we summarize the fundamental components of mucus secretion regulation, as depicted in Figure 3, to provide a fresh viewpoint for understanding the complicated function of mucus.

### 4.1. Host’s Immune-Dependent Regulation

Deepening the research on mucin has demonstrated that the synthesis and secretion of mucin is particularly susceptible to the cytokines, which originate from immune cells and are essential for crucial aberrant processes. Cytokines are principally categorized into two distinct classes depending on their mode of action: pro-inflammatory and anti-inflammatory cytokines. It is particularly intriguing that the levels of pro-inflammatory cytokines are part of the overall inflammatory milieu in an abnormal state, yet anti-inflammatory cytokines such as IL-10 and IL-4 regulate pro-inflammatory activity that can modify immune-mediated disorders such as those in the gut. It is worth noting that, depending on the inflammatory setting, some cytokines, including IL-6, IL-22, and IL-27, are recognized to be key pro-inflammatory or anti-inflammatory cytokines. Evidence from genome sequencing studies implies that JAK-STAT and the NF-κB- STATs signaling pathway play a vital role in achieving functional identity by acting as a transducer and activator of transcription from different cytokines, respectively. Extensive research has documented that the cytokine possesses the ability to regulate mucin synthesis and gene expression and contribute to the maintenance of cell homeostasis. 

Upon activation of the TNF-α pathway, it has been shown to either upregulate the transcription of *Muc2* through the PI3K/AKT/NF-κB pathways or downregulate through the JNK pathway in the IBD model [109], In contrast to the above result, a number of large cross-sectional studies further suggest certain constraints between the signal pathways of NF-κB and JNK, leading to *Muc2* transcription without any obvious influence by pro-inflammatory cytokines [109], including IL-1 and TNF-α, which has been found to stimulate the expression of *MUC2* mRNA on the intestinal cancer cell line LS180 [110]. Moreover, IFN-γ seems to be favorably controlled in Cl.16E to promote mucin exocytosis without affecting mucin gene expression [111,112]. In addition, IL-1β stimulated the *MUC2* and *MUC5A* mRNA expression by activating the p38/ERK pathways and CREB/ATF1 [110,113]. Separate research found that the Th1 cytokines TNF- and IFN- inhibit not only the formation of intestinal mucin but also the rate of mucin transport from the Golgi to the secretory vesicles during C. *rodentium* infection [114]. This may imply that the impact of Th1 cytokine on goblet cells is still indeterminate. Furthermore, type 2 cytokines are a major regulator of mucin gene expression in an in vivo model [115] and, more importantly, as IL-4 and IL-13 facilitated *MUC2* and *MUC5AC* transcription by activating either the STAT6 or NF-κB pathways with ovalbumin (OVA) in the asthma model [116,117]. Recent studies also greatly supported a direct interaction with U0126, a specific inhibitor of the mitogen-activated protein kinase (MAPK) pathway, completely inhibiting the IL-4- or IL-13-increased *MUC2* mRNA level in the epithelium, which provides direct evidence that the Erk1/2 and P38 MAPK signaling pathways are involved in LS174T and HT29 cells and OVA-challenged animals in vitro [118,119]. TNF-α and IL-6 can interfere with the expression of the genes encoding glycosyltransferase, lowering the overall glycosylation level of mucin [120]. Paul V. and colleagues found that IL-4/13 T helper 2 cytokines and RA upregulate core 2β1,6 n-acetylglucosamine transferase in the human airway epithelial cell lines, potentially leading to altered mucin carbohydrate structure and properties [121].

When pathogens invade the host, it is more probable that the host will mount an appropriate immune response. On the harmful side, numerous xenobiotics, including aflatoxin M1, ochratoxin A, trichothecene mycotoxins toxin, and deoxynivalenol, have demonstrated an impact on goblet cell activities and MUC2 expression while also being carcinogenic. Recent evidence has supported an increasingly crucial role for mucin in innate immune defenses against xenobiotics. Subsequent studies identified that transcriptional suppression is caused by several mechanisms, including IRE1/XBP1, protein kinase R, and P38 MAPK, as well as mitochondrial dysfunction [122]. Infection of *Giardia duodenalis* (assemblage A) may cause diarrhea, with enhanced *MUC2* gene expression in human colonic epithelial cells, via both the protease-activated receptor 2 (PAR_2_) that is activated by trypsin and giardia cysteine protease activity (CPs), by degrading chemokines; the above evidence was also confirmed though Ca^2+^ free and the inhibitor of the ERK1/2-MAPK cascade, which restrains the *Muc2* transcription profile [123].

The current evidence reveals that there are substantial linkages between mucin granule accumulation and mucus exocytosis, and the cellular processes are dependent on various intersecting cytokines and tissue-derived factors [124]. Furthermore, this evidence underlines the importance of the host’s immune dependence in the regulation of mucin secretions and the operation of the gut barrier. Only cytokines, however, do not interpret the host in order to govern the composition and/or function of mucus. Thus, neural control of mucus production has emerged as an inevitable issue.

### 4.2. Enteric Nervous System

The enteric nervous system (ENS) has an ability to accurately regulate gastrointestinal homeostasis through intricate neural and glial networks to gastrointestinal crypts, which contribute to maintain various physiological processes including mucus secretion control. Mounting evidence manifests that the gastrointestinal tract is constantly contacted with microbiota, which conducts multiple advantageous activities that protect host health. Recent studies have further depicted that submucosal neurons, which may be found near lamina propria immune cells (e.g., APCs and ILCs), are well-adapted to interact with epithelial cells and play an essential role in mucin production [125]. AChR activation on goblet cells, inducing mucin secretion, was for the dynamic regulation of the goblet cells to adapt to environmental changes [126].

Vasoactive intestinal peptide (VIP) is involved in regulating the number and function of intestinal goblet cells. Experimental evidence corroborated that VIP-deficient mice exhibited impaired goblet cell development, resulting in significantly decreased expression of MUC 2 and Tff 3 as well as overt intestinal barrier dysfunction, possibly due to decreased expression of Cdx2, the master regulator of intestinal function and homeostasis [127,128]. Similarly, VIP treatment ameliorates the phenotype of DSS colitis in VIP KO mice, while also promoting mucosal integrity via mucus secretion acceleration. Given the above description, mucosal immune responses may lead to a better knowledge of inflammation-related pathogenesis as well as novel therapeutic intervention strategies for intestinal barrier dysfunction. Due to various CNS disorders correlated with intestinal barrier dysfunction, we provide a fresh peek into the mechanisms that promote gut microbe–ENS interactions and also aid in the development of innovative treatment approach in clinical practice.

Besides the most typical T cells involved in IBD, IL-6 exerts influence on many other cells. In CD, the secretion of IL-6 in epithelial cells leads to the reduction in intercellular cell adhesion molecule-1 (ICAM1) via NF-κB [129,130]. Incubation with IL-6 results in less expression of the endothelial protein C receptor (EPCR) and thrombomodulin by the vascular endothelial cell (VEC), consequently enhancing coagulation and limiting the anti-inflammation effect [131]. This evidence hints that treatment targeting IL-6 may bring a promising prospect.

### 4.3. Diet Ingredients Impact on Mucin Secretion 

Based on the current published evidence, increased dietary fiber intake appears to be associated with not only both local and whole-body chronic inflammation but also enhancement of the diversity of colonic microbiota and the production of short-chain fatty acids. The effect of dietary fiber on mucus production is the focus of this section. Dietary fiber appears to improve barrier function by increasing the expression of goblet-cell-derived mucins as well as coordinating mucin exocytosis. The underlying mechanism, in terms of process engagement, shows complicated connections that link with different types of food. Evidence from a gnotobiotic model shows that dietary fiber can alleviate mucus degradation by reducing the expression of the CAZymes, sulfatases, and proteases of β-glucosidase as well as restraining pathogens’ adhesion, which directly attacks mucus polysaccharides or translocates, leading to inflammation and/or increased pathogen susceptibility; this was further confirmed under the condition of reduced dietary fiber [132]. Simultaneously, the transcription of the primary colonic *Muc2* was modestly enhanced, but not the transcription of *Muc5ac*, *Tff1*, *Tff3*, or *Klf3*, suggesting a possible compensatory response by the host to counteract the increased bacterial mucus degradation [133]. Furthermore, dietary fiber may not only bind to toxins and lower the activity but also act as a decoy for the bacteria that bind to protect the intestinal epithelial cells [134]. A high-protein diet increases the profile of mucus degradation while decreasing mucus thickness, which may be connected to the severity of colitis [135,136].

There is no doubt that the harm of a high-fat diet to the body has been widely accepted, but its effect on the intestinal mucosa remains a key signal that is currently overlooked. According to research, a mucin-type O-glycosylation structure can be caused by improper glycosylation and may lose a crucial role as a barrier effector due to a high-fat diet. In particular, Maria. M et al. found that the downregulation of mucin synthesis was lower, as was as the amount of the glycosylated residuals in Kras-mutant mice induced by a high-fat diet [137], which in turn caused susceptibility to inflammatory disease, increased tumor risk, and pathogens [138]; all of these can be reversed by capsaicin by involving TRPV1 to restore the gut barrier [42].

Currently, food additives, including emulsifiers, are widely used in the catering industry, and their role of a “double-edged sword” has attracted wide attention detailed in Box 4. Food additives are routinely added to a range of processed foods to enhance the appearance and extend the shelf life. They were initially classed as compounds that disrupt host–pathogenic microbe interaction patterns. A histomorphological investigation of mice given carboxymethylcellulose (CMC) or polysorbate-80 rather than water indicated a reduction in both the closeness of bacteria to the epithelium and the number of mucolytic bacteria such as *Ruminococcus gnavus* [139]. Furthermore, using Citrobacter rodentium (CR)-induced infectious colitis, treatment with Pectin and Tributyrin diets decreased colitis severity while renewing *Firmicutes* and *Bacteroidetes* and increasing mucus production [140]. Butyrate promotes cancer cell metastasis by secreting many molecules, including growth factors and other cytokines, and facilitates M2 macrophage (Mφ) polarization with elevated expressions of several well-known surface makers, including CD206. CD206, also known as C-type lectin, is expressed on the surface of Mφs and some subsets of the immature dendritic cells participating in antigen presentation. In light of the scientific evidence supporting the diverse health benefits of a balanced diet including dietary fiber, a high-fat diet and food additives are, therefore, areas for identifying effective preventive strategies for improvements in both metabolic and overall health.

Box 4Food additives as a magnifying glass for mucus-related diseases.A food additive ensures the needs of
quality, safety, or nutrition. There are hundreds of food additives derived
from plants, animals, or minerals or artificially synthesized in use today,
the majority of which have complex components and diverse functions [141]. The study into
the effects of food additives on various organ systems of the body is still in
its early stages. However, recent studies demonstrated that emulsifiers
exacerbate ileitis, disequilibrate symbiotic strains, and destroy disulfide
bonds of plycosylated protein; all these effects lead to intestinal barrier
malfunction, and ultimately result in inflammation or tumorigenesis. Indeed,
emerging evidence demonstrates that dietary additives may raise the risk of
metabolic diseases.

### 4.4. Microbial Colonization 

The gut microbe, being one of the most complex ecological communities, serves as the barometer of intestinal health with a double-edged-sword role. When intestinal microbes maintain a dynamic balance, they participate in the regulation of various functions of the body. However, when the balance is broken, the destructive effect of intestinal microbes on the body sparks widespread alarm. The emphasis here is on reviewing the existing knowledge on the influence of gut bacteria on the intestinal mucosa. By regulating the *Tff3* gene and affecting the production of a secretory peptide involved in the repair and regeneration of goblet cells [139], lactic acid bacteria were deemed probiotic, with barrier function integrity in vitro and in vivo. *Lactobacillus reuteri* JCM1081 is linked to neutral carbohydrate chains via a residue on the non-reducing terminal, as opposed to the acidic carbohydrate chains that contain sialic acid via the anchor sequence (LPXTG) located at the C-terminus. This process occurs at the sites of the specific interactions between *Lactobacillus* and the carbohydrate chains rather than the electrostatic force [142,143,144]. Consequently, the adhesion potential of *Lactobacillus* has received great attention, and it may be the most direct factor for the enhancement of the intestinal innate immune barrier. Due to the aforementioned evidence, this is also an essential development strategic objective for probiotics in the treatment of enteric-related diseases. 

Above all, understanding the interactions between *Lactobacillus* and mucin is crucial for elucidating the survival strategies of LAB in the GI tract. Of particular note, *Akkermansia Muciniphila* can not only degrade and utilize host mucus and the metabolites produced by the decomposing oligosaccharides in mucin, but it can also be used by a broader range of intestinal microorganisms to synthesize vitamin B_12_, 1, 2-propylene glycol, propionic acid, butyric acid, and other compounds that are beneficial to microflora and intestinal barrier homeostasis [145]. According to the aforementioned considerable evidence presented on intestinal microbes and mucin, this finding and the recognition of the mechanisms have deepened our understanding of the interaction between the mucin and health.

## 5. Mucus Abnormalities and Disease

The benefit from the status of the mucus and interactions with the regulatory system of the body determine human fitness and the underlying health status. Furthermore, these alterations in the biological barriers differ not only between diseases but also at the systemic, microenvironmental, and cellular levels, making them difficult to isolate and extensively characterize. In the enteritis model, for example, the permeability of the inner mucus layer rises, enabling the penetration of molecules or pathogens that could not pass through in a normal condition [41]. So we are going to use some diseases to further illustrate the relationship between mucus abnormalities and diseases (as depicted in Table 3).

### 5.1. Sjogren’s Syndrome and Mucin Deficiency 

Sjogren’s syndrome (SS) is a heterogeneous, multifactorial disease influenced by multiple factors such as genetic, environmental, and hormonal parameters, which include but are not limited to clinical and pathologic phenotypes from dry eye disease to chronic interstitial lung disease (ILD) to gut disease [175]. Despite the amount of research on SS, the underlying harm to various organs remains unknown [176]. Recently, some experimental evidence corroborated that not only does SS not only increases autophagy and death and diminishes goblet cell density but also reduces MEM formation via increased epithelial growth and adhesion molecule (ICAM)-1, human leukocyte antigen (HLA)-DR, and IL-6. Advanced culture-independent methods, such as next-generation sequencing, make it easier to analyze the microbial communities, the IFN-a that is derived from dendritic cells, and CD4^+^, which is one of the focuses of SS. In response to activating T cells, various cytokine secretion can be aggrandized, such as IFN-γ, IL-2, IL-6, IL-10, etc., which are linked to a multitude of the pathological processes and underlying mechanisms of SS [177]. Aside from the aberrant activation and proliferation of the B-cell, anti-Ro (SSA), and anti-La (SSB) auto-antibodies, which are a hallmark of SS, the short cytoplasmic RNP-bound peptides SSA-60kD, SSA-52kD, and SSB-48kD are too. However, studies suggest that the direct effects of anti-Ro52 auto-antibodies can cause glandular dysfunction in the SS group [178]. Indeed, abundant evidence implies that different immune components are active at various phases, that is, the condition is characterized by the aberrant mucus production caused by proinflammatory mediators [179,180] or the inflammatory edema of the conjunctiva via the activation of P2X7 [181,182].

### 5.2. Bidirectional Relation between Mucin and Cystic Fibrosis

Cystic fibrosis (CF), the most common autosomal recessive disease caused by mutations in the cystic fibrosis transmembrane conductance regulator (CFTR) gene, is a progressive and fatal lung disease, along with mucus-obstructive lung disease and sticky mucus as well as the recurrent lung infections accompanying pathological changes in the digestive organ. Excessive accumulation of sticky mucus activates mucilage-degrading enzymes and aberrant or excess degradation mucus, especially the glycans of gel-forming mucins, not only MUC5AC/B but also MUC2 and MUC6, which in turn directly impact the susceptibility to pathogens. Absent or reduced CFTR activity can lead to a decrease in chloride and bicarbonate transport, lowering sputum pH and perhaps increasing mucus density and sulfate residues. MUC5AC/B secretion increased during CF as did mucus viscoelasticity [183]. Many studies demonstrated that more sialylated and less sulfated airway-mucus qualities modify microbial features [156]. To sum up, we have grounds to assume that CF immune surveillance and immunotherapy offer a rationale for the treatment approach. 

### 5.3. Helicobacter Pylori and Mucin

The link between gastric carcinoma and its underlying mechanism has been subject to intense investigation since the discovery of Helicobacter pylori (*H. pylori*) in 1982. Due to *H. pylori* being the etiologic candidate of chronic gastritis, gastroduodenal ulcers, mucus-associated lymphoid tissue lymphoma, and gastric carcinoma, the gastric pathogen H. *pylori* has been regarded as a serious public health problem; indeed, some studies showed that *H. pylori* is directly related to the occurrence of gastric cancer. Due to aberrant mucin secretion, *H. pylori* infection causes the adhesion and persistence of the infection. A lot of evidence hints that during *H. pylori* infection, the abundant charge density (sialylation and sulfation) of gastric mucin oligosaccharides was frequently presented [184], suggesting that *H. pylori* infection adheres to gastric epithelial cells, causing the apoptosis and destruction of decarbohydrate structures. Particularly, the Lewis b structures have been identified as secreting MUC5AC mucin-related structures and as a potential binding site for bacterial adhesins. Lewis b is the major blood-group-related antigen seen in the stomach epithelium of persons in a positive-secretor status. According to several pieces of research, *H. pylori* infection is also linked to soluble MUC 1 mucin, the effect of which influences their interaction [185]. Interestingly, adhesin-binding sites of the mucin in BabA and SabA as well as low pH have a high affinity with *H. pylori*, leading to atrophic gastritis and eventually gastric carcinoma. Given the prior discussion, the major goal of this part is to explicitly indicate that improving the novel and more refined therapeutic method to eradicate *H. pylori* infection is needed [186].

### 5.4. IBS and IBD

IBS and IBD have been extensively explored in recent years as key actors in intestinal diseases [187,188], but their etiology remains poorly understood. The relationship between the mucus alteration in protein expression and quantity and the interaction of these diseases has not been elucidated. Aside from the typical abnormalities in the stomach, a similar feature implies that these two illnesses may be caused by joint processes. Some evidence from multiple levels of experimentation suggests that with MUC2 loss, diminished secretion and/or a change in carbohydrate composition might alter the start or pathogenesis of IBD/IBS. Additionally, MUC5AC/B and the MUC7 gene profile were aberrant in IBD/IBS [189,190], which may result in the increased permeability of mucus and the nocuousness of the intestinal barrier [191]. In contrast, other studies indicate that, aside from the enhancement of the activity and stability of immune cells, modification in the backbone of mucin-type O-glycans avoids unfavorable intestinal mucosal immune tolerance. There are several pieces of research on IBD/IBS, particularly in recent years, as well as numerous studies on the correlation between mucus changes and IBD/IBS, which are not described here.

### 5.5. Mucin in Cancer

Mucins not only protect interface tissues from pathogens, but they also restrict the activation of a critical connection between harmful inflammatory responses and cancer at the epithelial barrier. In contrast to other solid tumors, the tumor cells generated from epithelial cells actively manufacture and employ mucin to promote tumor development and invasion.

Mucins are crucial components of innate immunity due to the host’s response to pathogen infection. A recent study found that recurring chronic inflammation is a key risk factor that can contribute to tumor formation. In fact, excessively high levels of mucins on cancer cells, acting as a mask for tumor-associated antigens (TAAs), hinder both the specific and non-specific lysis of tumor cells. We hypothesize that this might be another underlying escape mechanism for tumor cells because it not only enhances cytokine production to degrade mucus barrier function but also attenuates tumor responsiveness to cytotoxic agents. Furthermore, MUC5B and MUC7 contribute to the chronic inflammatory state via Toll-like receptor 4 (TLR4) recognition, generating an accumulation of pro-inflammatory cells, which in turn drive injurious inflammatory signaling. Strong evidence demonstrates that the disturbance of polarity or cell–cell contacts (e.g., adherens junctions, tight junctions) is one of the most important elements in epithelial oncogenesis. Recent studies of TM (MUC1, MUC4) revealed that they play key roles in numerous types of cancer and bacterial infections, such as *H. pylori* [192]. Regardless, improper MUC1, MUC4, and MUC13 expression leads to abnormalities of the epithelial-mesenchymal transition (EMT) signaling pathways in the gastrointestinal tract as well as the mechanism by which aberrant catenin, ZEB1, ERK, and ERBB activation causes metastatic development. Thus, the disturbed interplay between TJ and cancer cells facilitates the motility and invasion of tumorigenic cells [193]. Despite extensive investigation, no conclusive agreement has been established on the function of signaling competent mucin, identified as a fundamental driver of cell proliferation, growth, and survival [194]. Thereby, overexpressed mucins cause diminution by the detoxification of the chemotherapeutic drug, which may portend a more favorable outcome for improved cancer chemotherapy efficacy by decreasing the secretion of mucus [195].

## 6. Conclusions

With the deepening of research in recent years, mucus no longer receives people’s attention as a dependent variable that changes with diseases, so more studies have discussed mucus and diseases as part an organic whole that also influence each other. In this paper, we firstly summarized the basic biological characteristics of mucus and mucin and the underlying mechanisms of synthesis, release, and regulation. Then, we also compared the dynamic correlation between the mucins and the diseases, as shown in Table 1 and Table 2. Lastly, we highlighted the crucial role digestive tract mucosa plays in maintaining the integrity of a healthy condition as well as potential targets for some disease challenges. Of course, the fundamental issue for many studies is that the mucin structure is complicated and diverse, while the synthesis process is governed by multiple factors. Thus, the study of the mucin structure is not particularly thorough at the moment. However, we have seen that scientists’ eyes are starting to pay more attention to this detail, so it is believed that the analysis and depicted summary of mucus in this review should draw more attention and generate fresh routes for future clinical practice and even future cancer-targeted treatment.

## Figures and Tables

**Figure 1 ijms-24-04227-f001:**
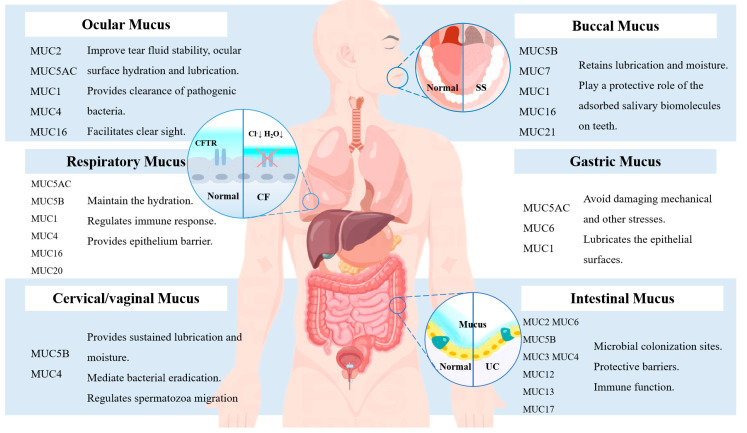
Diagram depicting a broad overview of mucus distribution, variation, and disorders linked with mucus throughout the body. The profit in the ocular cavity is mostly based on MUC1/2/5AC16, which has a primary role to lubricate and clean the mouth. Mucus in the respiratory tract is mostly MUC1/4/5B, and its primary function is to establish a barrier while also keeping the respiratory tract moist and participating in the immune response. The cervical/vaginal tract mucus is mostly MUC5 B/4, which has lubricating, antimicrobial, and sperm-motility-enhancing properties. The mucus in the oral cavity is mostly MUC7/6/21/5B, which has lubricating and moistening properties. The mucus in the stomach is primarily MUC5AC/6/1, which lubricates epithelial cells and prevents mechanical injury. Mucin in the intestine is primarily MUC6/5B/3/4/2/17, which serves as a protective barrier and provides microbial colonization.

**Figure 2 ijms-24-04227-f002:**
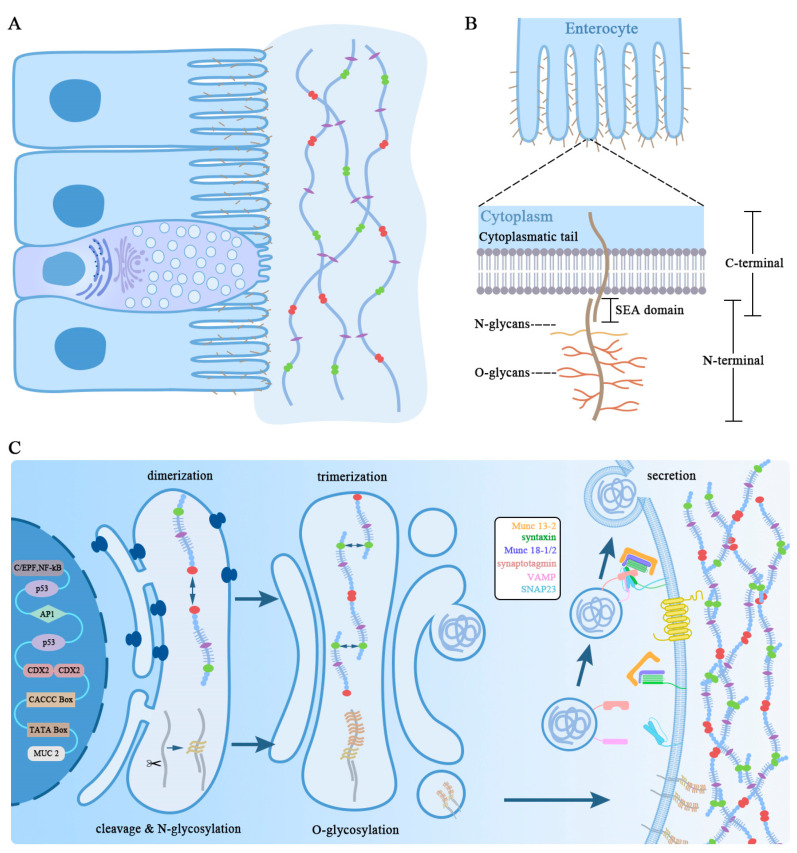
Schematic diagram of MUC2 synthesis, assembly, glycosylation, and process. (**A**) Patterns of distribution and structure of gel-forming secreted from GCs. (**B**) The basic skeleton of membrane-associated mucins shown in the mucin polypeptide chain is N-glycosylated and forms disulfifide-bonded dimers through its C-terminal Cys-rich domains. (**C**) Schematic representation of the dimers transported from ER to the Golgi compartments for the assembling process via a variety of intricate and ongoing processes. The large polymers are deposited in mucin granules in goblet cells and secreted by endocytosis.

**Figure 3 ijms-24-04227-f003:**
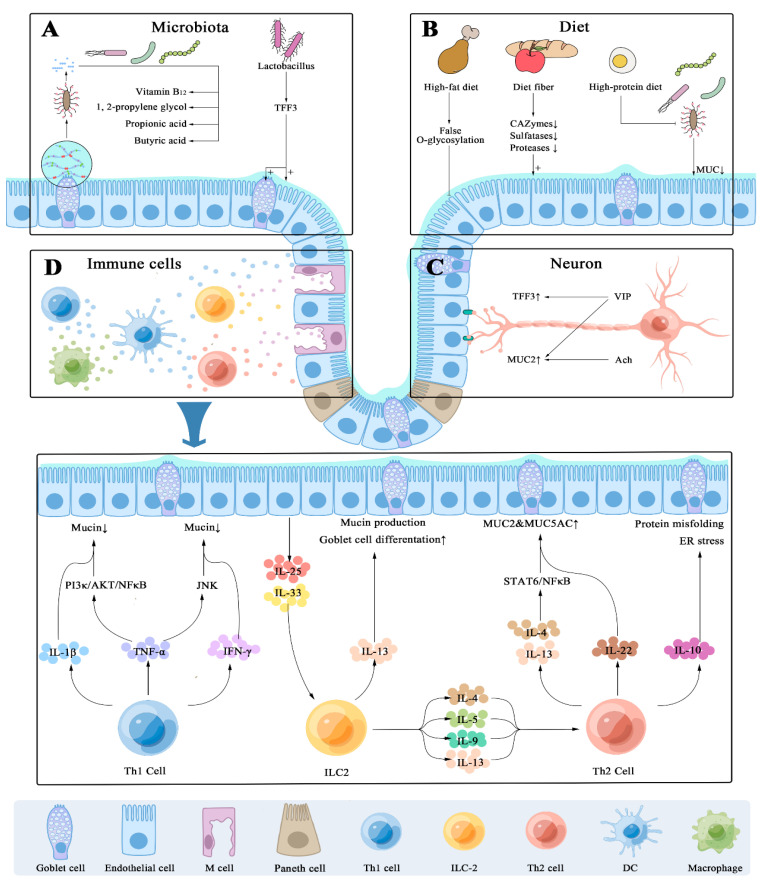
Schematic diagram showing possible factors influencing mucous secretion in the intestinal tract. The secretion of intestinal mucus is mediated by both external and internal factors. These components, such as the microbiome (**A**), food (**B**), neurotransmitters (**C**), and cytokines (**D**), all have a variety of effects on mucus synthesis and secretion.

**Table 3 ijms-24-04227-t003:** Illustration of mucin-related diseases.

Disease	Sjögren’s Syndrome	Cystic Fibrosis	*H. pylori* Infection	CD	UC	CRC
Main organs	Mouth, eye	Lungs, pancreas	Stomach	Small intestine	Small intestine	Colon, rectum
Cardinal symptoms	Dry accompanies other immune system disorders	Cough, infection, nutritional deficiencies	Pain, bloating, ulcers	Pain, diarrhea, rectal bleeding	Bloody purulent stool, abdominal pain or cramping	Bloody purulent stool, change in bowel habits, abdominal pain or cramping
Mucus thickness	↓	Sticky and thick	↓	↑	↓	↓
Glycosylation	↓	↑	↓	Unknown	↓	↓
Sialylation	↓	↑	↑	↑	↑	↑
MUC1	↑	↑	↑	↓	↑	↑
MUC2	?	↑	↑	↓	↓	↓
MUC3	?	↑	Undetectable	↓	Unchanged	Unchanged
MUC4	Undetectable/unchanged	↑	Undetectable	↓	↑	↓
MUC5AC	↑	↑	↓	↓	↑	↑
MUC5B	Unchanged	↑	↑	↓	↓	Unchanged
MUC6	?	↑	↑	↑	↑	Undetectable
MUC7	↑	?	Undetectable	?	?	Unchanged
MUC8	?	Unchanged	Undetectable	Undetectable	Undetectable	?
MUC10	?	?	?	Undetectable	Undetectable	?
MUC12	Undetectable	?	?	↓	↓	↓
MUC13	Undetectable	?	?	↑	↑	↑
MUC15	Unchanged	?	?	?	?	↑
MUC16	Unchanged	↑	Undetectable	?	?	↑
MUC17	Undetectable	?	?	↓	↓	?
MUC18	?	?	?	↑	↑	?
MUC19	↓	?	?	?	?	?
MUC20	↓	?	?	↓	↓	↑
MUC21	↓	?	?	?	?	?
Refs.	[146,147,148,149,150]	[151,152,153,154,155,156]	[157,158,159]	[160,161,162,163,164]	[160,161,162,163,165]	[71,166,167,168,169,170,171,172,173,174]

↑: increase. ↓: decrease. ?: no related report. Undetectable, the molecule was not detected. Unchanged, content level did not change. Unknown, the changing trends are complex and uncertain.

## Data Availability

Not applicable.

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
