# Peer review of "View from the Biological Property: Insight into the Functional Diversity and Complexity of the Gut Mucus"

_ijms, 2023, doi:10.3390/ijms24044227_

Round 1
Reviewer 1 Report
Recently, the attention of scientists has been focused on the relationship between the gut mucus and the intestinal function. Gut mucus can be useful in fulfilling a number of critical activities in the maintenance of cellular and organismal homeostasis, offering potential therapeutic targets for prevention in clinical practice.
The work submitted to me for review, entitled the " View from the biological property: Insight into the Functional Diversity and complexity of the gut mucus " concerns an interesting topic, which is the effect gut mucus on the protection of the intestinal and digestive function.
Abstract: we need a sentence reflecting the immune function of the mucus, I could not find anything related to that on the abstract. Please explain or rephrase.
Introduction: Line 53: the aim of the review is not clear, please illustrate it in detail. It is suggested to expand the introduction.
Line 146: MUC2, Please mention the full name in the beginning of the sentence.
Line 159: the same for IgA
Line 186: why the references are bold here?
Line 265: space before in
Line 307: space before different
Line 375: Please delete space before regulate
Line 379: Please delete space after that
Line 421: Please delete space after also
Line 440: Please delete the extra space
Line 456: please rewrite the reference 129-130 correctly.
Line 476: space before furthermore
Line 520: Please delete extra space after various
Line 583: please delete extra space
Conclusion: Please improve the conclusions of the summary.
Author Response
Recently, the attention of scientists has been focused on the relationship between the gut mucus and the intestinal function. Gut mucus can be useful in fulfilling a number of critical activities in the maintenance of cellular and organismal homeostasis, offering potential therapeutic targets for prevention in clinical practice.
The work submitted to me for review, entitled the " View from the biological property: Insight into the Functional Diversity and complexity of the gut mucus " concerns an interesting topic, which is the effect gut mucus on the protection of the intestinal and digestive function.
Abstract: we need a sentence reflecting the immune function of the mucus, I could not find anything related to that on the abstract. Please explain or rephrase.
Response: Sincerely thank you for your positive comments on our article, and also express our thankfulness for your pointing to the limitation shortcomings in this abstract. According to your sincere advice, we further make corresponding additions and adjustments to the important contents of the abstract, showing the functional relationship between mucus and gut. After the revision, we also think that this can highlight the important role of the theme of this article. The modified parts have been marked in blue. Thanks again. Please refer to page 1 line 34-37.
Introduction: Line 53: the aim of the review is not clear, please illustrate it in detail. It is suggested to expand the introduction.
Response: Thank you in particular for your careful reading and positive pointing out the shortcomings of this article, so according to the content of your constructive comments, we have revised the content of the abstract in this article, and we also believe that this setting can make the theme of the article more prominent. Thank you again for your sincere advice. The section where the changes were made has been highlighted in bright blue in the article. Thanks again for your suggestions. Please refer to page 2 line 79-82.
Line 146: MUC2, Please mention the full name in the beginning of the sentence.
Response: Thanks for your meaningful tips, based on your tips, we modified the corresponding part in the article again. Please see on the page 4 line 142.Thanks again.
Line 159: the same for IgA
Response: Thank you for your generous tip. According to your advise, we have modified the corresponding part again on page 5 line 171.Thanks again.
Line 186: why the references are bold here?
Response: Thank you very much for your important and constructive comments and for pointing out the inadequacies of our article. Based on your proposal, we have reconfirmed the references section and found that we have gained from the previous use, so we have added two more references here. Please see ref. 23 and 24 on page 5 line 199.Thank you again for your conscientious and responsible spirit.
Line 265: space before in
Response: Sincerely thank you for your careful and serious questions about the format of our article. According to your suggestions, we have removed the spaces in the corresponding sections. Please see on the page 7 line 277.
Line 307: space before different
Response: Thank you for carefully pointing out the shortcomings in the article. According to your generous suggestion, we have deleted the space. Please refer to the page 9 line 319. Thank you again.
Line 375: Please delete space before regulate
Response: Thank you very much for your suggestions on the shortcomings of our article. According to your suggestions, we have deleted the space. Please see on the page 12 line 387. Thanks.
Line 379: Please delete space after that
Response: Thank you very much for your suggestions on the shortcomings of our article. According to your suggestions, we have deleted the space in the relevant process. Please see on the page 13 line 391.Thanks.
Line 421: Please delete space after also
Response: Thank you very much for your kind suggestions on the limitation of our article. According to your advice, we have deleted the space. Please see on the page 14 line 433. Thanks.
Line 440: Please delete the extra space
Response: Thank you very much for your suggestions on the shortcomings of our article. According to your kind remider, we have deleted the redundant blank space in the relevant process. Please see on the page 13 line 452. Thanks.
Line 456: please rewrite the reference 129-130 correctly.
Response: Sincerely thank you for your careful cross-examination of the shortcomings of the references in the article. According to your suggestions, we have further modified the references here. Thank you again. Please see on the page 13 line 468.Thanks.
Line 476: space before furthermore
Response: Thank you for your carefulness and pointing out some flaws in our article. According to your suggestion, we have deleted the extra spaces in the article. Thank you again for your sincere and careful efforts. Please see on the page 15 line 488.Thanks again.
Line 520: Please delete extra space after various
Response: Thank you very much for your careful reading and finding out the shortcomings in the article. According to most of your suggestions, we have also delete extra space. Please see on the page 15 line 532.Thanks.
Line 583: please delete extra space
Response: Thank you very much for your kind reminder on the shortcomings of our article. According to your suggestions, we have deleted the space on the page 18 line 595.Thanks.
Conclusion: Please improve the conclusions of the summary.
Response: Thank you in particular for your sincere proposal on the conclusion part of this article. According to your hints, we have made further careful analysis and judgment on the conclusion part, and revised the content of this part to make its conclusion part more precise and concise. Thank you again for your sincere suggestions. Modifications in this section have been highlighted in bright blue. Please see on the page20-21 line 675-688. To express my sincere thanks for your generous recommendations.

Reviewer 2 Report
This review manuscript is well organized and written how there are some minor comments:
1. There are a few grammatical and spelling errors
2. The Figures and Tables need legends for some of the important abbreviations to make the manuscript readable to a wider audience.
3. Also, there are a lot of abbreviations in the text that are not explained, which will be challenging for some readers
Author Response
This review manuscript is well organized and written how there are some minor comments:
- There are a few grammatical and spelling errors
Response: Sincerely thank you for your positive evaluation on the manuscript , which also makes our work more confident. At the same time, thank you for your generous suggestions on the limitations of this article. According to your kind suggestions, we have read the full text carefully again, and corrected some grammatical and spelling errors in the article. All corrections have been marked in bright blue in the text. Once again, I would like to express my heartfelt thanks for your constructive suggestions.
- The Figures and Tables need legends for some of the important abbreviations to make the manuscript readable to a wider audience.
Response: Sincerely thank you for your thoughtful suggestions. Due to your kind suggestions, we have further improved the important information in the chart in the article, and marked some of his abbreviations. Thank you again for your sincere advice. The revised parts of the article have been marked in blue. Please refer to page 21 line 689.
- Also, there are a lot of abbreviations in the text that are not explained, which will be challenging for some readers.
Response: Thank you very much for your generous comments. As you said, there are many abbreviations in the article, which may reduce the readability of some readers. According to your suggestion, we have further standardized the writing of the full name of the first abbreviation in the article. In this way, the readability of our article is better. All revised parts have been marked in bright blue in the article. Once again, I would like to express my heartfelt thanks for your sincere suggestions. Please refer to page 21 line 691.

Reviewer 3 Report
The abstract should be a total of about 200 words maximum.
Add a space along the text between word and reference in brackets
Figure 1: add in the legend the meaning of all MUC
Line 87: add a brief paragraph relating to absorbable cells and secretable cells
Line 195: add a paragraph in which the components (water, lipids, electrolytes, proteins and others) that make up the mucus are analyzed and how they vary in the various traits, in physiological and pathological states
Line 679: in the abbreviations delete one anti-microbial proteins
Author Response
- The abstract should be a total of about 200 words maximum.
Response: Thank you very much for your constructive comments. According to your suggestions, we have further condensed and refined the abstract part of the article, making the content of the abstract more accurate, so now the number of words in the abstract has reached 198.Please see the summary section below. Please refer to the article on page 1 line 14-27. Thanks.
Due to mucin important protective effect on epithelial tissue, it has garnered extensive attention. The role played by mucus in the digestive tract is undeniable. On the one hand, mucus forms “biofilm” structures that insulate harmful substances from direct contact with epithelial cells. On the other hand, a variety of immune molecules in mucus also play a crucial role in the immune regulation of the digestive tract. Due to the enormous number of microorganisms in the gut, the biological properties of mucus and its protective actions are more complicated. Numerous research has hinted that aberrant expression of intestinal mucus is closely related to impaired intestinal function. Therefore, this purposeful review is to give highlight of the biological characteristics and functional categorization of mucus synthesis and secretion. In addition, we have highlighted a variety of regulatory factors for mucus. Most importantly, we also summarize some changes and possible molecular mechanisms of mucus during certain disease processes. All these are benefit to clinical, diagnosis and treatment and can provide some potential theoretical basis. Admittedly, there are still some deficiencies or contradictory results in the current research on mucus, but none of this diminished the importance of mucus in protective impacts.
- Add a space along the text between word and reference in brackets
Response: Thank you from the bottom of my heart for your kind advice. According to your suggestion, we have added a space between the text and the reference in the article, which seems to make the reference and the text clearer. There are a lot of modifications in the article, so we can’t mark them here one by one, but thank you again for your suggestions.
- Figure 1: add in the legend the meaning of all MUC
Response: Sincerely thank you for your suggestions and pointing out the shortcomings of Figure 1 of our article. According to your suggestions, we have added its explanatory part to the unification of the article. I think this addition and change will be very helpful to you for a correct and comprehensive understanding. Thank you again for your suggestions. Please refer to the article on page 2 line 75-83.
Figure1. Diagram depicting abroad overview of mucus distribution, variation, and disorders linked with mucus throughout the body. The profit in the ocular cavity is mostly based on MUC1/2/5AC16, which primary role is to lubricate and clean the mouth. Mucus in the respiratory tract is mostly MUC1/4/5B, and its primary function is to establish a barrier while also keeping the respiratory tract moist and participating in the immune response. The cervical/vaginal tract mucus is mostly MUC5 B/4, which has lubricating, antimicrobial, and sperm motility enhancing properties. The mucus in the oral cavity is mostly MUC7/6/21/5B, which has lubricant and moistening properties. The mucus in the stomach primarily serves as MUC5AC/6/1 by lubricating epithelial cells and preventing mechanical injury. Mucin in the intestine is primarily MUC6/5B/3/4/2/17, which serves as a protective barrier and provides microbial colonication.
- Line 87: add a brief paragraph relating to absorbable cells and secretable cells
Response: Sincerely thank you for your constructive and profound suggestions on this issue. According to your suggestions, we have added a box to this topic, so that it can be used as a more independent topic to explain this issue. We think this may be more appropriate. We also sincerely thank you for your valuable suggestions. Please refer to the box 1 on page 3 line 91-98.
Box 1 Brief review of the differences between absorbable nuclei and secretory cells
Wrinkles or folds cover absorbable cells structurally. Microvilli are finger-like projections on individual epithelial cells. The plicae circulars, villi, and microvilli serve to enhance the amount of surface area accessible for nutritional absorption. Secretable cells can be characterized in addition to their glandular form by the manner of secretion and the type of chemicals emitted. The secretions are encased in vesicles that migrate to the cell's apical surface, where they are released by exocytosis. Saliva containing the glycoprotein mucin, for example, is a merocrine secretion
- Line 195: add a paragraph in which the components (water, lipids, electrolytes, proteins and
others) that make up the mucus are analyzed and how they vary in the various traits, in physiological and pathological states
Response: Thank you very much for your insight proposal. According to your generous hints, the relevant content has been added to the article to make the description of this part more novel. Therefore, we have further revised and improved the content of this part. The content is as follows. Please refer on page 5 line 205-214. Thank you again for your wise advice.
The viscoelastic secretion produced by goblet or mucous producing cells and found on the epithelial surfaces of all organs accessible to the outside world. Mucus is a complex aqueous fluid having viscoelastic, lubricating, and hydrating qualities due to the glycoprotein mucin in conjunction with electrolytes, lipids, and other smaller proteins. Under physiological circumstances. However, mucin secreted under pathological conditions has multiple structural differences, and its limitations affect the normal physiological role of mucus, resulting in some disease-related changes, such as an alteration in its dense cellular permeability, which allows pathogenic microorganisms and poisons to cause direct stimulation of epithelial cells. A great number of studies have shown that immune escape reactions of tumor cells are common in tumors when the structure of the mucus changes.
- Line 679: in the abbreviations delete one anti-microbial proteins
Response: Thanks for your kind opinion, we have removed anti-microbial proteins from the acronym based on your suggestion. Please refer to the abbreviations on page 21 line 699. Thanks.
